# AutoGraph: Enabling Visual Context via Graph Alignment in Open-Domain Multi-Modal Dialogue Generation

Deji Zhao*
Northeastern University
School of Computer
Science and Engineering
Shenyang, China
zhaodeji@stumail.neu.edu.cn

Donghong Han†
Northeastern University
School of Computer
Science and Engineering
Shenyang, China
handonghong@cse.neu.edu.cn

Ye Yuan
Beijing Institute of
Technology
School of Computer
Science and Technology
Beijing, China
yuan-ye@bit.edu.cn

Bo Ning
Dalian Maritime University
School of Information
Science and Technology
Dalian, China
ningbo@dlmu.edu.cn

Mengxiang Li
China Telecom Corp Ltd
Beijing, China
limengx@126.com

Zhongjiang He
China Telecom Corp Ltd
Institute of Artificial
Intelligence (TeleAI)
Beijing, China
hezj@chinatelecom.cn

Shuangyong Song
China Telecom Corp Ltd
Institute of Artificial
Intelligence (TeleAI)
Beijing, China
songshy@chinatelecom.cn

## Abstract

Open-domain multi-modal dialogue system heavily relies on visual information to generate contextually relevant responses. The existing open-domain multi-modal dialog generation methods ignore the complementary relationship between multiple modalities, and are difficult to integrate with LLMs. To tackle these challenges, we introduce AutoGraph, an innovative method for constructing visual context graphs automatically. We aim to structure complex information and seamlessly integrate it with large language models (LLMs), aligning information from multiple modalities at both semantic and structural levels. Specifically, we fully connect the text graphs and scene graphs, and then trim unnecessary edges via LLMs to automatically construct a visual context graph. Next, we design several graph sampling grammar for the first time to convert graph structures into sequence which is suitable for LLMs. Finally, we propose a two-stage fine-tuning strategy to allow LLMs to understand graph sampling grammar and generate responses. We validate our proposed method on text-based LLMs, and visual-based LLMs, respectively. Experimental results show that our proposed method achieves state-of-the-art performance on multiple public datasets.

## CCS Concepts

• **Computing methodologies** → **Natural language generation**; **Information extraction**; • **Information systems** → **Multimedia information systems**.

*Work done during internship at TeleAI.
†Corresponding author.

## Keywords

Multi-modal alignment, Dialogue graph, Dialogue generation

**ACM Reference Format:**
Deji Zhao, Donghong Han, Ye Yuan, Bo Ning, Mengxiang Li, Zhongjiang He, and Shuangyong Song. 2024. AutoGraph: Enabling Visual Context via Graph Alignment in Open-Domain Multi-Modal Dialogue Generation. In *Proceedings of the 32nd ACM International Conference on Multimedia (MM '24), October 28–November 1, 2024, Melbourne, VIC, Australia.* ACM, New York, NY, USA, 10 pages. https://doi.org/10.1145/3664647.3681012

## 1 Introduction

Open-domain multi-modal dialogue generation has garnered increased attention in recent years due to its ability to closely mimic real-life scenarios and generate contextually appropriate responses [8, 30]. Dialogue systems are no longer limited to textual forms, and visual information plays a crucial role in dialogue agent. Multimodal dialogue systems can comprehend not only textual but also other modal information to generate appropriate responses. This integration of multi-modal information into traditional text-based dialogue systems, known as open-domain multimodal dialogue systems, has attracted increasing research interest [22, 32, 35, 36, 52].

Unlike previous Visual Question Answering (VQA) tasks [1] that focus on a single, or small number of images related to the context, the open-domain multi-modal dialog generation task has a lot third-person viewpoint multi-modal information. Although existing models have shown promising performances, they still suffer from two problems. Firstly, recent image-grounded dialogue models [17, 33, 45] endeavor to enhance dialogue generation capabilities by integrating relevant images into the dialogue models. Video-grounded dialogue models [7, 19, 20] aim to combine video modal information with dialogue systems. These methods encode multi-modal infromation through different encoders respectively, while ignoring the complementary relationship between the different modalities. As shown in Figure 1, if the visual contextual information is ignored, it is difficult to clarify the specific coreference relationship between 'this', 'it' and 'you'. Encoding visual

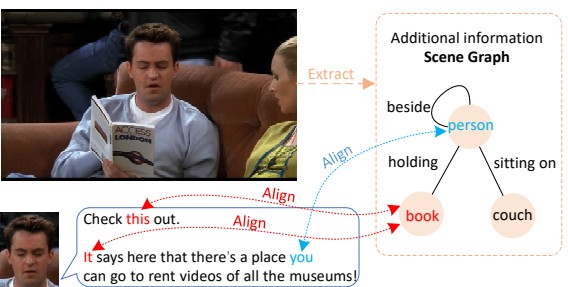

**Figure 1: An utterance extracted from the MELD dataset. It is insufficient only consider text information in open-domain multi-modal dialogue generation. There should be interaction between different modalities. Using scene graphs can serve as an expansion of the dialogue, filling in crucial information.**

context information separately involves a lot of noise. And it is difficult to map information from multiple modalities to a unified multi-modal vector space, which leads to the lack of factual and incoherent responses.

Secondly, the multi-modal community has sparked new interest in enhancing Large Language Models (LLMs) with visual information [14], with representative models such as LlaVA [20], VisCPM [7], and Monkey [16]. But existing image-grounded or video-grounded models' features are difficult to combine with LLMs due to the different vector spaces, which limits the ability of LLMs to uniformly model multiple modal features. It has been demonstrated that leveraging scene graphs can effectively enhance the understanding of image modalities [51]. The GlaMM model [31] try to push LLMs to generate scene graphs sentence to understand of the image. Structure-CLIP proposed by Huang et al. [9] employ scene graph knowledge to fit image-text matching tasks via multi-modal language models. However, these methods focus on VQA task and still struggle to handle multi-turn conversations or video modal information with a large number of frames present. The GraphGPT [39] model attempts to integrate text-based graph structures with LLMs and takes the index of nodes in the graph as direct input to the context. However, GraphGPT's comprehension ability is poor for graph structures with numerous nodes because LLMs cannot accurately establish the correspondence between indexes and nodes.

To address the aforementioned two issues, text is used as a cue to align multiple modalities in this paper. LlaVA's [20] approach inspires the idea that we can use images to augment text-based LLMs. To establish complementary relationships between different modalities and incorporate them with LLMs, we design an automatically constructed multi-modal context graphs method and several graph sampling grammar, called AutoGraph. We aim to structure complex information and seamlessly integrate it with large language models (LLMs), aligning information from multiple modalities at both semantic and structural levels. The AutoGraph method is a general approach that can enhance the visual capabilities of LLMs.

More specifically, in order to obtain aligned multi-modal context graphs, we employ the semantic dependency graph parsing to

extract the structure of the utterance and obtain a textual graph. And the scene graph parsing method is used to convert the video modalities into image graphs. We fully connect the text graph and image graphs to form a holistic heterogeneous graph, then design a pruning strategy to align the graph structures of the two modalities at the semantic level. In order to combine the aligned multi-modal graph with LLMs, we devise three types of graph sampling method to transform the graph structures into a sequential structure. We argue that while the shift to sequentialization may change the word order, for LLMs it amounts to learning a new grammar to establish a mapping relationship with the target responses. During the fine-tuning stage, we propose a two-stage fine-tuning strategy to enable the LLMs to better comprehend the graph sampling grammar we designed. We validate our proposed method on text-based LLMs, and visual-based LLMs, respectively. Experimental results show that our proposed AutoGraph method can effectively enhance the performance of different types of LLMs and achieve the best results on multiple public datasets.

The main contributions are summarized as follows:

- We align multiple modal information at the semantic and structural levels through an automatically constructed visual context graph.
- For the first time, we propose three effective graph sampling grammar for transforming graph structures into sequences, seamlessly integrating the aligned visual context graph with LLMs. And we introduce a two-stage fine-tuning strategy to enhance the understanding of graph sampling grammar by LLMs.
- We conduct experiments via text-based LLMs and visual-based LLMs as backbone models. Experiments on two public datasets demonstrate that LLMs augmented by our proposed AutoGraph method exhibit superior visual dialogue capabilities.

## 2 Related Work

### 2.1 Open-domain Dialogue Generation

Open-domain multi-modal dialog generation is a task that relies heavily on understanding the different modal context. Open-domain multi-modal dialogue generation can be divided into two categories: image-grounded and video-grounded.

*2.1.1 Image-grounded approaches.* Image-grounded approaches integrate image-based visual information into dialogue systems. Open-domain dialogue datasets based on image-grounded include MMChat [52], DialogCC [13], MMDialog [4], Image-Chat [34], and others. Maria [17] and VisAD [33] model proposed by Liang et al. and Shen et al. respectively, attempt to integrate contextually relevant images with dialogue systems. Tu et al. [41] explicitly categorize visual knowledge into finer granularity turn-level and entity-level, proposing the RESEE model to incorporate visual representations into dialogue models through modality concatenations. Zhang et al. [46] propose the ZRIGF model, which includes contrastive and generation pretraining modules, mapping different modalities to the same vector space and generating appropriate responses. VisCPM [7] and Monkey [16] are Multi-modal Large Language models (MLLMs), which are pre-trained by image and

context information. These MLLMs establish mapping relationships between multiple modalities through extensive data.

*2.1.2 Video-grounded approaches.* Video-grounded dialogue can be seen as a more complex extension of image-grounded dialogue, as videos contain numerous frames that drastically increase computational complexity. Open-domain dialogue datasets for video-grounded dialogue include MELD [26], OpenViDial [22], OpenViDial 2.0 [43], Tiktalk [19], and others. Lin et al. [19] convert videos into multiple frames and use an image encoder to encode video modal information. They concatenate the different modalities' feature and feed them into a decoder to generate responses. However, these methods overlook the potential complementarity and enhancement between different modalities.

## 2.2 Graph Structure for Dialogue and Fusion with LLMs

*2.2.1 Graph Structure for Dialogue.* The advantage of applying graph structure in dialogue is that it can simulate the flow of information. DialogueGCN [5] is proposed by Ghosal et al. for dialogue emotion recognition tasks. Peng et al. [24] construct a dialogue graph to highlight both global and local features of the conversation. Kim et al. and Zhang et al. [11, 47] improve the performance of dialogue systems through coreference relationships across multiple modalities. Zhao et al. [50] construct dialogue graphs to model the speaker's cognitive shifts during the conversation. A expand strategy is proposed by Zhao et al. [49] to enlarge the constructed dialogue graph. But the dialogue graph constructed by aforementioned methods is based on utterance level, the AutoGraph model in this paper is focus on words level.

*2.2.2 Fusion with LLMs.* Incorporating structured information into LLMs can effectively enhance the model's performance. ERNIE 3.0 [38] simply converts graph triplets into a tokenized text passage as input. K-BERT [21] injects knowledge triplets into sentences through visibility matrices to reduce the sequence length, with only knowledge entities being included as part of the sequence. To further distill knowledge, CoLAKE [37] proposes a unified word knowledge graph, where tokens from input sentences form a fully connected word graph. GraphGPT [39] directly uses the index of nodes in the graph as input to the context window, enabling LLMs to comprehend the graph's topology. However, these methods are merely sequentialization approaches for different knowledge triplets, rather than a method for converting a large connected dialogue graph into a sequence. Our proposed AutoGraph method aimd to design a graph sampling grammar to reduce the knowledge noise and seamlessly integrate context graph with LLMs.

## 3 Method

### 3.1 Task Definition

The goal of the open-domain multi-modal dialog generation task is to generate appropriate responses based on contextual information with multi-modality. We formulate the this task as follows. Given text modal $ContextsT = \{U_1, U_2, ..., U_n\}$ and visual modal $ContextsV = \{V_1, V_2, ..., V_n\}$, $n$ is the number of turns in a dialogue. $V_i$ and $U_i$ are videos and utterances at the $i$th turn. The target response is $Y = (y_1, y_2, ..., y_m)$, where $m$ is the number of words.

### 3.2 Overview of the Architecture

In this paper, we propose the AutoGraph model to automatically comprehend and align text and video modal information, the structure of the model is shown in Figure 2. The AutoGraph model consists of three modules, §3.3 Visual Context Graph Construction, §3.4 Graph Sampling Grammar and §3.5 Two-stage Fine-tuning. We start with semantic dependency graph parsing and scene graph parsing to obtain structured relations from text and images, respectively. Then, we fully connect the text graph and the scene graph, and let LLMs trim meaningless edges to obtain the Visual Context Graph via the Few-shot and In-Context Semantic Alignment Prompt. We design graph sampling grammar to transform graph structures into sequence structures, and finally use a two-stage fine-tuning strategy to prompt the LLMs to understand this new graph sampling grammar.

### 3.3 Visual Context Graph Construction

To align textual and video modalities in visual context, we construct a structured visual context graph based on semantic alignment. The visual context graph $\mathcal{G}^{VT}$ is a heterogeneous graph, consisting of the text graph $\mathcal{G}^T$ and the image graph $\mathcal{G}^V$.

$$\mathcal{G}^{VT} = \{\mathcal{G}^T; \mathcal{G}^V\} \quad (1)$$

It is worth mentioning that our proposed AutoGraph approach focuses on the word level rather than the sentence level. We believe that achieving better alignment with other modalities and constructing the visual context graph requires more fine-grained knowledge. In the field of context graph construction, there has been a lot of work. In the MuSE model, Zhao et al. [49] split sentences of speakers using punctuation marks to build the context graph. Zhao et al. [50] individually constructed dialogue context graphs from the perspective of different speakers. These graphs focus on the sentence level, treating the utterances of speakers as nodes in the graph.

*3.3.1 Text to Graph.* In the part of text-to-graph transformation, as highlighted in the green box in Figure 2, we use semantic dependency graph parsing [2] to obtain the dependency graph between words. We split the speaker's context $ContextsT$ into multi-turns $\{U_1, U_2, ..., U_n\}$ and then separately parse the semantic dependency graph for each utterance $u_i$, obtaining the text graph $\mathcal{G}^{T_i}$, where $i$ represents the $i$th turn of the context.

$$\mathcal{G}^{T_i} = DependencyParsing(U_i) \quad (2)$$

Taking $U_1$ in Figure 2 as an example, the result after parsing with the semantic dependency graph is shown in Figure 3. In Figure 3, the edges between words represent some form of dependency relationship in the semantic dependency graph parsing. However, to avoid introducing redundant information, these relationships are not used in this paper, and only the edges are employed as connections between nodes. Finally, we get the text graph $\mathcal{G}^T = \{\mathcal{G}^{T_1}, \mathcal{G}^{T_2}, ..., \mathcal{G}^{T_n}\}$, where $n$ is the number of turns.

*3.3.2 Image to Graph.* Due to the high frame rate of video modality, pre-training based multi-modal approaches for extracting image features face challenges in meeting the demands of processing multiple frames of images. Following the previous approach to

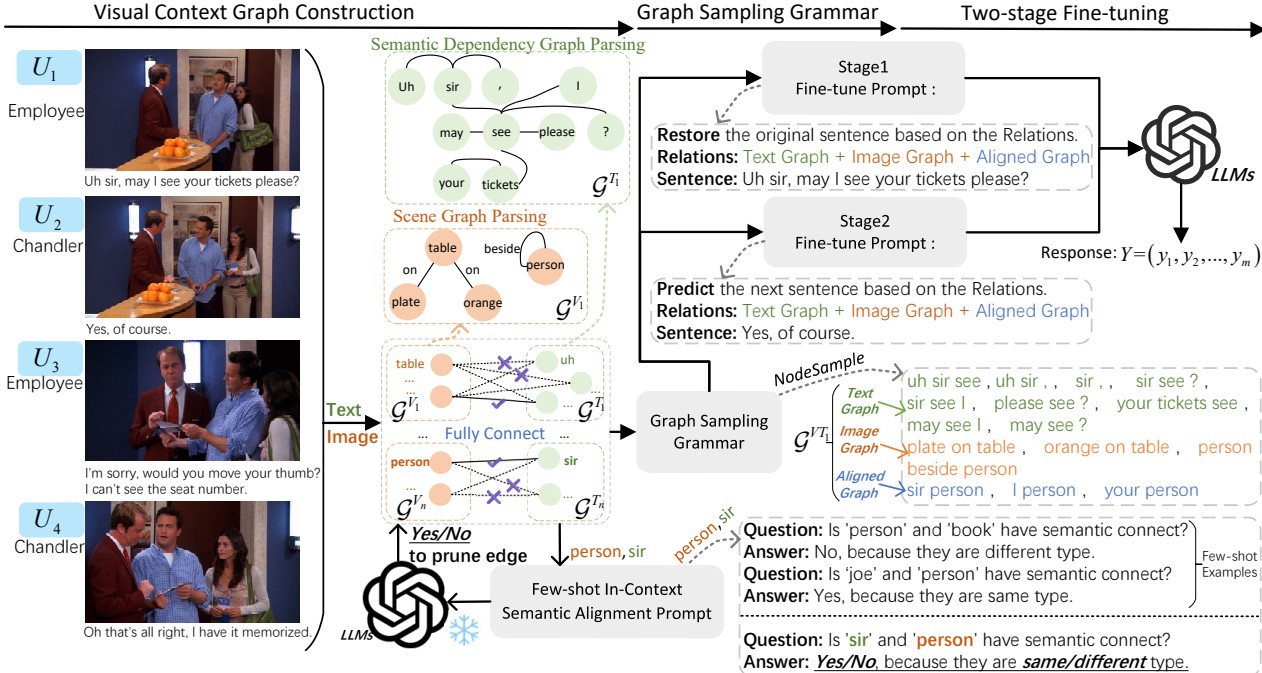

**Figure 2: The architecture of our proposed AutoGraph model, which consists of three modules: §3.3 Visual Context Graph Construction, §3.4 Graph Sampling Grammar and §3.5 Two-stage Fine-tuning.**

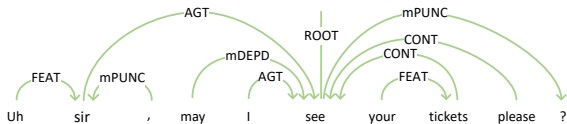

**Figure 3: Example of semantic dependency graph parsing of $U_1$ in Figure 2. Relationships between words are not used in AutoGraph model.**

extract keyframes from video [19, 22, 43], we extract each video clip $V_i$ into multiple keyframes $f_j$.

$$f_j = KeyFrameExtract(V_i), \quad (3)$$

$$V_i = \{f_1, f_2, ..., f_j\}, \quad (4)$$

$j$ is the number of total key frames in video clip $V_i$.

For each keyframe $f_j$, we employ scene graph parsing [44] to construct the scene graph at the $i$th turn.

$$\mathcal{G}_j = SceneGraphParsing(f_j) \quad (5)$$

The graphs of multiple keyframes are merged as subgraphs to form the image graph $\mathcal{G}^{V_i}$.

$$\mathcal{G}^{V_i} = \{\mathcal{G}_{f_1}, \mathcal{G}_{f_2}, ..., \mathcal{G}_{f_j}\}, \quad (6)$$

$j$ is the number of total frames in video clip $V_i$. Finally, the same method is applied to construct image graph $\mathcal{G}^V$ for each video clip and $\mathcal{G}^V = \{\mathcal{G}^{V_1}, \mathcal{G}^{V_2}, ..., \mathcal{G}^{V_n}\}$.

*3.3.3 Graph Alignment.* To align multiple modalities at the word level, we first **fully connect** the text graph $\mathcal{G}^T$ and the image graph $\mathcal{G}^V$. Next, a semantically prompt for LLMs is designed to guide the LLMs to prune edges between nodes of different modalities, resulting in the aligned visual context graph $\mathcal{G}^{VT}$. We employ the Few-shot In-Context Learning method [42] to trim meaningless edges. Taking the $U_1$ in Figure 2 as an example, the Few-shot In-Context Semantic Alignment Prompts are shown as follows.

**Few-shot In-Context Semantic Alignment Prompt:** "Question: Is 'person' and 'book' have semantic connect? Answer: No, because they are different type. Question: Is 'joe' and 'person' have semantic connect? Answer: Yes, because they are same type. Question: Is 'word A' and 'word B' have semantic connect? Answer: ". The **word A** and **word B** are two nodes (words) from textual and image graph. The first two questions and answers serve as examples for LLMs to reference.

After traversing both ends of all fully connected edges between **word A** and **word B**, we finally obtain the aligned visual context graph $\mathcal{G}^{VT}$.

$$\mathcal{G}^{VT} = \{\mathcal{G}^{VT_1}, \mathcal{G}^{VT_2}, ..., \mathcal{G}^{VT_n}\}. \quad (7)$$

### 3.4 Graph Sampling Grammar

At present, most LLMs can only draw knowledge from sequential structures and cannot directly understand graph-based structure information. Inspired by structured knowledge-enhanced pre-trained models [21, 37–39], we attempt to transform the visual context graph $\mathcal{G}^{VT}$ into a sequential structure, allowing us to sample the topology of the graph. We design three different graph

sampling grammar based on graph-level sampling and node-level sampling, which are §3.4.1 *GraphSample*, §3.4.2 *NodeSample* and §3.4.3 *DeepNodeSample*.

*3.4.1 GraphSample.* In graph-level sampling, similar to the node feature update mechanism in Graph Convolutional Networks (GCN) [12], for the target node $N_i$, the *GraphSample* function can sample all neighboring nodes within $1 - hop$ (including $N_i$).

The graph-level sampling sequence *GraphSample* of the entire visual context graph $\mathcal{G}^{VT}$ can be calculated as follows:

$$GraphSample = \bigcup_{i=0}^{M}\bigcup_{j=0}^{M} A_{ij}N_j, \qquad (8)$$

where $M$ represents the number of nodes and $A$ represents the adjacency matrix.

*3.4.2 NodeSample.* In the graph-level sampling, all neighboring nodes of the target node $N_i$ are sampled, resulting in many nodes being repeatedly sampled after traversing the entire graph $\mathcal{G}^{VT}$. Inspired by GraphSAGE [6], we modify the sampling grammar that only $K$ neighboring nodes of the target node $N_i$ will be sampled in node-level sampling. If the target node does not have $K$ neighboring nodes, we will not pad to $K$ nodes.

Building upon this, we sample first-order and second-order neighbor nodes to sample graph topology to a sequence. The node-level sampling sequence *NodeSample* of the entire visual context graph $\mathcal{G}^{VT}$ can be calculated as follows:

$$NodeSample = \bigcup_{i=0}^{M}\bigcup_{j=0}^{K} A_{ij}N_j \cup A_{ij}^2 N_j, \qquad (9)$$

$$A_{ij}^2 = \begin{cases} 0 & \text{if} \quad i = j \\ \min(1, A_{ij}^2) & otherwise \end{cases}, \qquad (10)$$

where $A^2$ represents the square of the matrix $A$ and $M$ represents the number of nodes. To avoid redundant sampling of the target node, we set the $A_{ij}^2 = 0$ in the adjacency matrix when $i = j$, and to prevent repeated sampling of other nodes, we set the values of the remaining nodes in $A_{ij}^2$ to 1 .

*3.4.3 DeepNodeSample.* Based on *NodeSample*, we further explore deep node sampling methods. We continuously sample first, second, and third-order neighbors of the target node $N_i$. The deep node-level sampling sequence *DeepNodeSample* of the entire visual context graph $\mathcal{G}^{VT}$ is calculated as follows:

$$DeepNodeSample = \bigcup_{i=0}^{M}\bigcup_{j=0}^{K} A_{ij}N_j \cup A_{ij}^2 N_j \cup A_{ij}^3 N_j, \qquad (11)$$

$$A_{ij}^3 = \begin{cases} 0 & \text{if} \quad i = j \\ \min(1, A_{ij}^3) & otherwise \end{cases} \qquad (12)$$

where $A^3$ represents the cube of the matrix $A$, $A^2$ is calculated via equation 10, $K$ represents the $K$ neighbors of target node $N_i$ and $M$ represents the number of nodes.

We do not explore deeper-level sampling as it would significantly increase the length of the graph sampling sequence. To the best of our knowledge, this is the first attempt to devise a grammar to facilitate LLMs in acquiring graph structure information. Given

**Table 1: Data statistics for the MELD dataset and OpenViDial dataset.**

| | MELD | OpenViDial |
|---|---|---|
| Train | 9989 | 974803 |
| Valid | 1109 | 55679 |
| Test | 2610 | 55667 |

that the LLMs have the ability to learn other forms of grammar [10], we believe that LLMs can also handle this type of graph sampling grammar.

## 3.5 Two-stage Fine-tuning

In order to facilitate the understanding of entirely new grammar structures by the LLMs, we devise a two-stage fine-tuning strategy.

The objective of the first stage is to enable the LLMs to comprehend this grammar, restoring utterance $U_i$ via the visual context graph $\mathcal{G}^{VT_i}$ of the $i$th turn. Through prompting, the LLMs reconstruct the sentence under graph sampling grammar into the original sentence, establishing a connection between the new grammar and human-familiar grammar. The goal of the second stage is to enable the LLMs to predict the response to the next utterance $U_{i+1}$ based on the visual context graph $\mathcal{G}^{VT} = \{\mathcal{G}^{VT_1}, \mathcal{G}^{VT_2}, ..., \mathcal{G}^{VT_i}\}$.

Taking the $U_1$ in Figure 2 as an example and two-stage fine-tuning prompt is shown as follows. The **Relations** in stage 1 and stage 2 is fixed, and the **Sentence** is the target sentence that the model needs to predict and participates in the loss calculation during the fine-tuning process.

**Stage 1 Fine-tuning Prompt:** "**Restore** the original sentence based on the Relations. **Relations**: uh sir see, uh sir „ sir „ sir see ?, sir see I, please see ?, your tickets see, may see I, may see ?, plate on table, orange on table, person beside person, sir person, I person , your person. **Sentence**: uh sir, may I see your tickets please? ".

**Stage 2 Fine-tuning Prompt:** "**Predict** the next sentence based on the Relations. **Relations**: uh sir see, uh sir, , sir „ sir see ?, sir see I, please see ?, your tickets see, may see I, may see ?, plate on table, orange on table, person beside person, sir person, I person, your person. **Sentence**: yes, of course. ".

## 4 Experiment

### 4.1 Datasets

We conduct experiments on two publicly available datasets, the MELD dataset [26] and the large-scale OpenViDial dataset [22]. The dataset statistics are summarized in Table 1. To stay close to open-domain conversation scenarios, we chose these two datasets. OpenViDial has a larger scale compared to the MELD dataset.

### 4.2 Baseline Models

- **MMChat** [52]: A multi-modal dialogue model based on a multi-layer text encoder, Fast-RCNN encoder and GPT-2 decoder.
- **RESEE** [41]: RESEE enhances the text-based conversational abilities via visual knowledge. The visual and text encoders are composed of CLIP [28] and T5 [29], respectively.

**Table 2: Results of automated evaluation on the MELD dataset (%) via Full fine-tuning approach.**

| Model | Distinct-1 ↑ | Distinct-2 ↑ | BLEU ↑ | Rouge-L ↑ | F-BERT ↑ | CHRF ↑ |
|---|---|---|---|---|---|---|
| RESEE | 4.8510 | 15.2295 | 0.4130 | 4.3863 | 75.2614 | 7.6507 |
| MMChat (GPT-2) | 4.6747 | 14.6951 | 0.3323 | 4.6110 | 74.7785 | 8.7579 |
| Llama2-7B (Text only LLMs) | 3.1687 | 10.0073 | 0.1304 | 2.2689 | 77.3152 | 7.6362 |
| LLaVA2-7B (Multi-modal LLMs) | 7.7715 | 22.8793 | 0.9121 | 5.9174 | 78.1487 | 8.3218 |
| Llama2-7B + AutoGraph (*GraphSample*) | 5.6010 | 21.0152 | 0.8493 | 3.6452 | 78.6225 | 7.1252 |
| Llama2-7B + AutoGraph (*NodeSample*) | **7.5920** | **34.8087** | **1.4942** | **6.3176** | 79.3841 | **11.6212** |
| Llama2-7B + AutoGraph (*DeepNodeSample*) | 6.8226 | 26.4739 | 0.6655 | 6.2572 | 79.2667 | 9.8351 |
| LLaVA-7B + AutoGraph (*GraphSample*) | 7.5184 | 24.9317 | 1.1178 | 5.3264 | 79.2583 | 10.1217 |
| LLaVA-7B + AutoGraph (*NodeSample*) | **8.3218** | **35.1276** | **1.5102** | **7.0311** | **79.4041** | 11.3278 |
| LLaVA-7B + AutoGraph (*DeepNodeSample*) | 8.1357 | 33.1671 | 1.3014 | 6.9912 | 79.3302 | **12.9315** |

**Table 3: Results of automated evaluation on the OpenViDial dataset (%) via Full fine-tuning approach.**

| Model | Distinct-1 ↑ | Distinct-2 ↑ | BLEU ↑ | Rouge-L ↑ | F-BERT ↑ | CHRF ↑ |
|---|---|---|---|---|---|---|
| RESEE | 2.0103 | 3.7157 | 0.6101 | 0.4538 | 76.3321 | 2.4317 |
| MMChat (GPT-2) | 1.9131 | 3.5038 | 0.5968 | 0.4151 | 75.8312 | 2.3945 |
| Llama2-7B (Text only LLMs) | 3.0734 | 6.9061 | 0.1092 | 0.1621 | 77.7771 | 2.2540 |
| LLaVA-7B (Multi-modal LLMs) | 3.9715 | 6.3417 | 0.5211 | 0.5248 | 78.2326 | 3.3493 |
| Llama2-7B + AutoGraph (*GraphSample*) | 2.8379 | 9.2816 | 0.5141 | 0.6501 | 78.4735 | 4.5733 |
| Llama2-7B + AutoGraph (*NodeSample*) | **4.0888** | **12.1607** | 0.6891 | 0.6091 | 78.4420 | 4.7637 |
| Llama2-7B + AutoGraph (*DeepNodeSample*) | 3.2510 | 8.5148 | **0.7278** | 0.6320 | **78.8388** | **6.0662** |
| LLaVA-7B + AutoGraph (*GraphSample*) | 3.6234 | 10.6582 | 0.6039 | 0.4670 | 78.3098 | 4.7296 |
| LLaVA-7B + AutoGraph (*NodeSample*) | **4.7310** | **13.4519** | 0.7421 | 0.4764 | 79.0943 | 5.2319 |
| LLaVA-7B + AutoGraph (*DeepNodeSample*) | 4.0101 | 13.1492 | **0.8602** | **0.6487** | **79.2304** | **6.6529** |

- **Llama2-7B** [40]: An open-source, high-performance, text only large language model for English. We fine-tune it using the textual modality of the MELD and OpenViDial datasets.
- **LLaVA-7B** [20]: The large language model based on Llama2-7B, fine-tuned by visual instructions, achieved excellent performance in several multi-modal tasks. After fine-tuning with the MELD and OpenViDial dataset, we use it as the baseline model.

## 4.3 Experiment Settings

We reproduce the results based on the source code provided in the original paper. All experiments are trained with the same parameters. The learning rate of QLoRA and Full fine-tuning is 1e-4 and 1e-5, respectively. All baseline models based on the LLMs are available through the corresponding open source projects. QLoRA rank is set to 128. In the AutoGraph model, $K$ is set to 3, and $M$ depends on how many neighbors the target node has. Experiments on the MELD dataset are accelerated by 8 * NVIDIA 32GB V100 GPUs, and experiments on the OpenViDial dataset are accelerated by 4 * NVIDIA 40GB A100 GPUs. Our code can be found through https://github.com/DericZhao/AutoGraph.

## 4.4 Evaluate Metrics

*4.4.1 Automatic Evaluation.* Automatic evaluation is efficient and fair. Following previous work, we adopt mainstream evaluation metrics, which include Distinct-$n$ [15], BLEU [23, 27], Rouge-L [18], F-BERT [48] and CHRF [25]. The Distinct-$n$ aims to encourage the dialogue system to generate diverse responses, while the BLEU, Rouge-L, F-BERT and CHRF seek to produce contextually relevant responses through different approaches.

*4.4.2 Human Evaluation.* Human evaluation is to evaluate responses quality from a human perspective. We conduct the aspect-based pairwise preference test for human evaluation. (1) Coherence (**Coh.**): which measures the relevance and coherence of the generated responses to the context. (2) Informativeness (**Inf.**): which response conveys more information related to context. (3) Grammar (**Gra.**): which measures whether the grammar of responses is correct. We randomly sample 200 response pairs of each model and recruit 10 evaluators to judge.

## 5 Results and Analysis

## 5.1 Automatic Evaluation Results

With AutoGraph method, different LLMs achieve state-of-the-art automatic evaluation results. We evaluate the effectiveness of our AutoGraph method on both the MELD dataset and the OpenViDial dataset. The results of automatic evaluation are shown in Tables 2, 3, 4, and 5. We employ QLoRA [3] and Full fine-tuning methods to fine-tune various LLMs.

Our experimental objectives include three main goals: **1.** Validate the performance of the AutoGraph method on the Llama2 text-based LLMs. **2.** Validate the performance of the AutoGraph method on the LLaVA multi-modal LLMs. **3.** Verify the sampling capability of

**Table 4: Results of automated evaluation on the MELD dataset (%) via QLoRA fine-tuning approach.**

| Model | Distinct-1 ↑ | Distinct-2 ↑ | BLEU ↑ | Rouge-L ↑ | F-BERT ↑ | CHRF ↑ |
|---|---|---|---|---|---|---|
| RESEE | 4.8510 | 15.2295 | 0.4130 | 4.3863 | 75.2614 | 7.6507 |
| MMChat (GPT-2) | 4.6747 | 14.6951 | 0.3323 | 4.6110 | 74.7785 | 8.7579 |
| Llama2-7B (Text only LLMs) | 2.8622 | 8.9845 | 0.1297 | 2.2876 | 76.0338 | 7.5832 |
| LLaVA-7B (Multi-modal LLMs) | 5.8526 | 23.3487 | 0.7621 | 5.3247 | 79.0131 | 7.9987 |
| Llama2-7B + AutoGraph (*GraphSample*) | 3.5183 | 14.4367 | 0.6896 | **6.3466** | 79.0821 | 8.0001 |
| Llama2-7B + AutoGraph (*NodeSample*) | **5.7986** | 24.4724 | 0.9366 | 6.1219 | 79.1248 | 8.8163 |
| Llama2-7B + AutoGraph (*DeepNodeSample*) | 5.5103 | 22.5282 | **0.9403** | 5.5804 | **79.2403** | 9.4888 |
| LLaVA-7B + AutoGraph (*GraphSample*) | 6.3241 | 23.2184 | 0.9872 | 7.0312 | 79.1342 | 9.3418 |
| LLaVA-7B + AutoGraph (*NodeSample*) | **7.8915** | **26.3398** | **1.4586** | **7.4596** | 79.2197 | 9.8149 |
| LLaVA-7B + AutoGraph (*DeepNodeSample*) | 7.5178 | 25.1284 | 1.0893 | 7.1574 | **79.2513** | **9.9324** |

**Table 5: Results of automated evaluation on the OpenViDial dataset (%) via QLoRA fine-tuning approach.**

| Model | Distinct-1 ↑ | Distinct-2 ↑ | BLEU ↑ | Rouge-L ↑ | F-BERT ↑ | CHRF ↑ |
|---|---|---|---|---|---|---|
| RESEE | 2.0103 | 3.7157 | 0.6101 | 0.4538 | 76.3321 | 2.4317 |
| MMChat (GPT-2) | 1.9131 | 3.5038 | 0.5968 | 0.4151 | 75.8312 | 2.3945 |
| Llama2-7B (Text only LLMs) | 2.1449 | 5.2219 | 0.0910 | 0.1803 | 77.7377 | 2.2325 |
| LLaVA-7B (Multi-modal LLMs) | 3.1285 | 7.0711 | 0.4598 | 0.2419 | 78.5334 | 3.4853 |
| Llama2-7B + AutoGraph (*GraphSample*) | 2.8486 | 7.7452 | 0.4786 | 0.2134 | 77.8720 | 2.4513 |
| Llama2-7B + AutoGraph (*NodeSample*) | **4.2391** | **11.8794** | 0.7251 | 0.2716 | 77.9364 | 2.9878 |
| Llama2-7B + AutoGraph (*DeepNodeSample*) | 4.1870 | 10.7114 | **0.8754** | **0.4926** | **78.6103** | **4.8728** |
| LLaVA-7B + AutoGraph (*GraphSample*) | 3.4517 | 9.2154 | 0.5128 | 0.4438 | 78.8111 | 3.5147 |
| LLaVA-7B + AutoGraph (*NodeSample*) | **4.5218** | **13.2411** | 0.7493 | 0.4507 | 78.9002 | 3.5574 |
| LLaVA-7B + AutoGraph (*DeepNodeSample*) | 4.3401 | 13.0013 | **0.8015** | **0.5042** | **79.1871** | **4.9210** |

the AutoGraph method with different graph sampling grammar for visual context graphs.

**1. AutoGraph on Text-based LLMs.** Compared to MMChat and RESEE models, which are specifically designed for open-domain dialogue generation, the Llama2-7B model with the AutoGraph method outperforms the baseline model across various metrics. The original text-based Llama2-7B model's performance do not surpass that of the MMChat model, but with the support of the AutoGraph method, the text-based model can also perform as well as multi-modal LLMs. This indicates that leveraging the AutoGraph method enables the text-based LLMs with the capability to comprehend visual scenes and then generate more appropriate responses in the open-domain dialogue generation task.

Compared to other general multi-modal LLMs, leveraging the best sampling methods in AutoGraph can enable the text-based Llama2-7B model to outperform the majority of baseline models. This demonstrates that employing the AutoGraph method can effectively enable text-based LLMs to comprehend multi-modal information and enhance model performance.

**2. AutoGraph on Multi-modal LLMs.** The LLaVA-7B model has been fine-tuned with visual prompts, and after fine-tuning on the MELD dataset and OpenViDial dataset, its performance surpasses that of models specifically designed for open-domain dialogue generation, such as MMChat and RESEE. We also try to incorporate the AutoGraph method into the LlaVA-7B model to enhance the understanding of scene information. Experimental results indicate that the model's performance is further improved after adding the visual context graph.

Compared to other general LLMs, LLaVA-7B exhibits superior performance after being enhanced by the AutoGraph method. Since LLaVA-7B already possesses visual capabilities, we believe that the AutoGraph method further provides the complementary relationship information. The experimental results indicate that our proposed AutoGraph method can also enhance multi-modal LLMs.

**3. Graph Sampling Ability.** We verify the sampling capabilities of different graph sampling methods for constructed visual context graphs. The *GraphSample* graph sampling method shows minimal enhancement in model performance. We attribute this to the noise caused by a significant number of duplicated nodes when traversing the entire visual context graph. The *NodeSample* graph sampling method alleviates the issue of repeated sampling. Models enhanced by the *NodeSample* sampling method tend to perform better on diversity metrics, such as Distinct-1 and Distinct-2. We believe this may stem from the insufficient understanding of the overall topology of the graph, as only $K$ nodes around the target node are sampled, potentially leading to incomplete sampling. The *DeepNodeSample* graph sampling comprehends the topology of the graph by sampling nodes at deeper levels. Models with the *DeepNodeSample* graph sampling grammar exhibit excellent performance on context relevance metrics, such as BLEU, Rouge-L, F-BERT, and CHRF scores.

Moreover, different fine-tuning methods also lead to different results. The experimental results of Full fine-tuning are significantly better than those of QLoRA. QLoRA is an efficient fine-tuning method as it only adjusts some layers' parameters in LLMs, while Full fine-tuning makes adjustments to all parameters. Both

**Table 6: Results of ablation experiments. Based on Llama2-7B, we eliminate the first-stage restore prompt, and only retain the second-stage predict prompt. All ablation experiments employ the Full fine-tuning method.**

| Dataset | Model | Distinct-1 | Distinct-2 | BLEU | Rouge-L | F-BERT | CHRF |
|---|---|---|---|---|---|---|---|
| MELD | w/o Stage1: Restore *GraphSample* | ↓ 0.3653 | ↓ 5.9942 | ↓ 0.0851 | ↓ 0.9468 | ↓ 0.2688 | ↓ 0.3524 |
| | w/o Stage1: Restore *NodeSample* | ↓ 1.1980 | ↓ 12.5266 | ↓ 0.1022 | ↓ 3.2360 | ↓ 0.4772 | ↓ 5.3403 |
| | w/o Stage1: Restore *DeepNodeSample* | ↓ 0.9837 | ↓ 6.3151 | ↓ 0.0831 | ↓ 2.8842 | ↓ 0.5791 | ↓ 4.1618 |
| OpenViDial | w/o Stage1: Restore *GraphSample* | ↓ 0.1880 | ↓ 0.4295 | ↓ 0.0476 | ↓ 0.0243 | ↓ 0.2730 | ↓ 0.3709 |
| | w/o Stage1: Restore *NodeSample* | ↓ 0.3339 | ↓ 1.5294 | ↓ 0.0829 | ↓ 0.0989 | ↓ 0.1583 | ↓ 0.5740 |
| | w/o Stage1: Restore *DeepNodeSample* | ↓ 0.1453 | ↓ 1.4099 | ↓ 0.0886 | ↓ 0.0671 | ↓ 0.2831 | ↓ 0.9278 |

**Table 7: Human evaluation results on MELD dataset (%). Ties are not shown. ‡represent significant improvement with $p$-value < 0.05. We select the *NodeSample* grammar with the best sampling capability as the model for manual evaluation.**

| Comparisons | Aspects | Win | Lose |
|---|---|---|---|
| Llama2-7B + | Coh. | **62.3**‡ | 26.1 |
| AutoGraph (*NodeSample*) **vs.** | Gra. | **65.1**‡ | 18.5 |
| MMChat (GPT-2) | Inf. | **68.8**‡ | 26.1 |
| Llama2-7B + | Coh. | **61.2**‡ | 25.5 |
| AutoGraph (*NodeSample*) **vs.** | Gra. | **62.4**‡ | 19.7 |
| Llama2-7B | Inf. | **60.5**‡ | 20.6 |
| LlaVA-7B + | Coh. | **51.5**‡ | 37.7 |
| AutoGraph (*NodeSample*) **vs.** | Gra. | **46.8**‡ | 35.4 |
| LLaVA-7B | Inf. | **49.7**‡ | 33.9 |

fine-tuning methods enable LLMs to comprehend the graph sampling grammar proposed in AutoGraph and adapt to two-stage fine-tuning.

## 5.2 Human Evaluation

In human evaluation, we are more concerned with whether the proposed graph sampling grammar can effectively map to the standard grammar in real-life. The human evaluation data on the MELD dataset is presented in Table 7. We evaluate the grammar of the response sentences, and LLMs demonstrate a full understanding of this graph sampling grammar from a human perspective. In terms of coherence and informativeness, the Llama2-7B model enhanced with the AutoGraph method has surpassed the text-based Llama2-7B model. In the human evaluation of the multi-modal LlaVA-7B model, there is a certain performance improvement after incorporating the AutoGraph method. Human evaluations further demonstrate that the AutoGraph method effectively enhances the visual dialogue capabilities of LLMs.

## 5.3 Ablation Experiments

We conduct ablation studies to verify the effectiveness of the two-stage fine-tuning strategy, as shown in Table 6. The results indicate that, after eliminating the first-stage restore prompts, the model's performance is affected to varying degrees. This indicates that the two-stage fine-tuning can assist the model in initially understanding the new grammar and providing support during the predict prompt process in the second stage. And the performance of LLMs with only predict prompt still surpasses most baseline models. Furthermore,

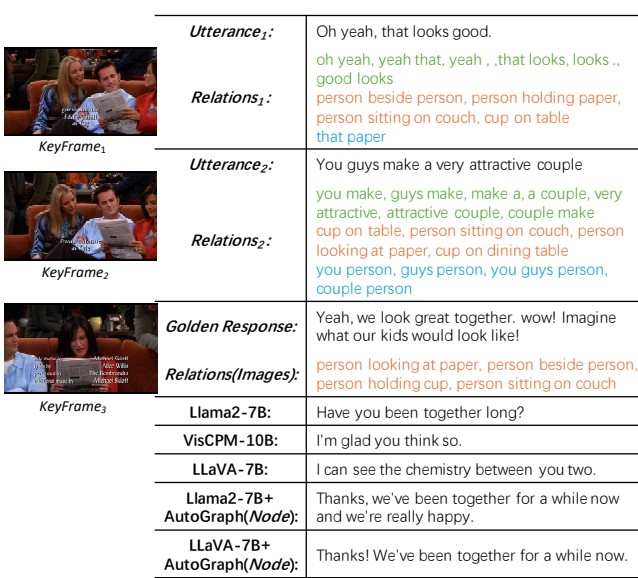

**Figure 4: Case study of the generated response. Text in orange is the scene graph relations extracted from images, text in green is the semantic dependency graph parsing relations and text in blue is the aligned relations.**

the most significant performance change is observed on the MELD dataset, which we attribute to its smaller data size compared to the OpenViDial dataset. Figure 4 shows the case study of different models.

## 6 Conclusion

In this paper, we propose an automated method for constructing visual context graphs to address the task of open-domain multi-modal dialogue generation. We first construct the visual context graph based on semantic and structural alignment. Then, to integrate the visual context graph with LLMs, we design several graph sampling grammar. Finally, we propose a two-stage fine-tuning strategy to enable the LLMs to comprehend the new grammar and generate responses. The experiments on text-based and multi-modal large language models validate the effectiveness of the AutoGraph method. In the future, we will explore how to automatically construct better dialogue graphs to integrate more structured information and develop dynamic graph sampling methods based on different edge weights.

## Acknowledgments

This work is supported by the National Natural Science Foundation of China (61672144, 61872072).

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
