# OpenReview forum: "AutoGraph: Enabling Visual Context via Graph Alignment in Open Domain Multi-Modal Dialogue Generation"
_acmmm.org/ACMMM/2024/Conference — MM2024 Poster_

### Official Review · Reviewer_asem · 2024-05-14

**Rating:** 3
**Confidence:** 4

**Summary:**

This paper aims to integrate LLM and existing open-domain multi-modal dialog generation techniques for deep relationship utilization. It designs AutoGraph method where the visual context graph aligns multi-modal information at multiple levels and it is transformed into sequences to use LLM.

**Strengths:**

1.	The analysis of the motivation is clear.
2.	The experiments are well organized and the case study is interesting with some inspiring observations.

**Limitations:**

1.	The task definition is unclear, is the target response supposed to match the visual modal context of i+1th turn? If so, V_n+1 should be given as input. The concepts mentioned in subsections of chapter3 do not correspond very well with the Figure 2, making it hard to understand. It would be better to adjust the structure of Figure 2, and present a more comprehensive case in the appendix.
2.	The experiments only compare with original LLM methods but not some dialog methods using LLM.
3.	The text graph is constructed using parsing tool, yet the parsing labels of the edges are not used, then how could the LLM know what the remaining empty edges means? It is quite unconvincing.
4.	Is the “Semantic Alignment Prompt” mentioned in the text applicable to all samples? If so, according to this prompt, any concepts of the same type will have connections, what is the basis for this? For instance, if there are only one personal pronoun and multiple names in the sentence, it is obviously not possible to connect the correct reference.
5.	The scene graph parsing focuses solely on nouns, would this result in a significant amount of image information being overlooked? For example, attributes like color, size, etc., and actions between items.
6.	It is difficult to understand which graphs are directed and which are undirected based solely on the provided figures and explanations, and the arrows are drawn somewhat casually. This will result in the misunderstanding of the subsequent graph operation.

**Suitability:**

3

---

### Official Review · Reviewer_cff8 · 2024-05-22

**Rating:** 5
**Confidence:** 3

**Summary:**

The paper introduces "AutoGraph," a method aimed at enhancing open-domain multi-modal dialogue systems by constructing visual context graphs. These graphs align semantic and structural information from multiple modalities (text and images/video), integrating this complex data with large language models (LLMs) to improve dialogue response quality. The technique involves creating connections between text and scene graphs, trimming irrelevant connections through LLMs, and converting these graphs into a sequential format that LLMs can process. The authors validate their approach through experiments with text-based and visual-based LLMs, showing improved performance on public datasets.

**Strengths:**

Innovative Integration: The paper effectively addresses the challenge of integrating multimodal data with LLMs by introducing a novel method for constructing and utilizing visual context graphs.

Detailed Methodology: The methodological approach, including graph sampling and fine-tuning strategies, is well-articulated, offering clear insights into the process and its novelty.

Robust Validation: The model is tested across different datasets and compared with several baselines, providing a thorough evaluation of its performance.

**Limitations:**

In this paper, although three distinct graph sampling strategies (GraphSample, NodeSample, DeepNodeSample) are proposed, each addressing specific issues independently, there is a lack of consideration for combining these strategies, which may limit flexibility and efficiency when handling complex graph structures. A Mixture of Sample (MoS) approach could potentially optimize model performance by integrating the advantages of each strategy, such as through hierarchical sampling or conditional sampling adapted to different graph structures and characteristics. The absence of such an integrated approach could be seen as a limitation of the current method. Future research could explore how to effectively combine these sampling methods to further enhance the model's generalization ability and manage complexity.

**Suitability:**

3

---

### Official Review · Reviewer_PLP4 · 2024-05-24

**Rating:** 3
**Confidence:** 2

**Summary:**

The paper presents AutoGraph, a method that automatically constructs visual context graphs, addressing the challenge of integrating complementary multi-modal information in open-domain dialogue systems. By enhancing LLMs with graph alignment, it improves response relevance and coherence, achieving state-of-the-art performance across various datasets, demonstrating efficacy in both text-based and multi-modal LLMs.

**Strengths:**

Compared to existing approaches, AutoGraph demonstrates superior performance, effectively enhancing both text-based and visually equipped LLMs, as evidenced through rigorous experimentation and notable improvements on diverse public datasets.

**Limitations:**

1.Unfortunately, the paper suffers from numerous grammatical errors and typos that detract from the clarity and professionalism of the research presentation.

2.While the study introduces and evaluates three different graph sampling methods, the results show that DeepNodeSample, a third-order node sampling technique, outperforms lower-order methods in terms of contextual relevance.  So why not explore higher-order sampling methods?

3.Why the authors decide to employ an older models such as GPT-2? Since the more advanced models like GPT-3.5 or GPT-4 is also available.

4. In Eq.6, how are multiple subgraphs fused, in a fully connected form?

**Suitability:**

2

---

### Official Review · Reviewer_GVcn · 2024-05-25

**Rating:** 2
**Confidence:** 4

**Summary:**

This paper proposes a novel approach for creating visual context graphs to enhance open-domain multi-modal dialogue generation. It can first construct the visual context graph, then perform graph sampling operations, and finally generate suitable responses through two-stage fine-tuning. Experimental results on multiple datasets validate the effectiveness of the scheme.

**Strengths:**

1.The structure of the article is clear.
2.The motivation is sound.

**Limitations:**

1.LLMs and graph networks tend to be less efficient. Does the fact that the authors have used both structures lead to extremely slow reasoning and thus no practical application?
2.The whole process is somewhat cumbersome and seems to be finely engineered. This can limit the generalizability and usefulness of the model.
3.The whole work is somewhat weakly innovative, it feels like a stacking of some existing work, like scene graphs + graph sampling + prompt engineering.
4.Many experimental details are not given, such as hyperparameter settings (K, M, etc.).
5.Several works in related fields should be cited, e.g.,
[1] Multimodal Dialog System: Relational Graph-based Context-aware Question Understanding.
[2] Structured Co-reference Graph Attention for Video-grounded Dialogue.

**Suitability:**

3

---

### Meta-Review · Area_Chair_rMtA · 2024-06-30

**Recommendation:** Accept (Poster)
**Confidence:** 2

**Metareview:**

This paper introduces a method for generating visual context graphs to improve open-domain multi-modal dialogue generation, demonstrating effectiveness through graph construction, sampling, and two-stage fine-tuning validated by experiments on various datasets.

Reasons to accept:
The motivation is clear and sound with corresponding method design.
Experiments are well organized with good performance.
Some core concerns have been addressed in the rebuttal, e.g., time efficiency and scalability.

Reasons to reject:
Multiple reviewers still have concern on innovation and usefulness (Reviewer GVcn, cff8).
Reviewers have concern on the paper presentation, including writing (Reviewer PLP4), core method introduction (Reviewer asem), and experiment details (Reviewer GVcn).
Reviewers ask for comparisons with dialog methods using LLM (Reviewer asem).

The paper receives the initial rating of weak reject, two borderline reject, and weak accept. After rebuttal and discussions, reviewers raise the rating to three borderline reject, and weak accept. After reading the paper, review comments, and discussion, AC recommends accept, and strongly encourages authors to incorporate the content from the rebuttal into the final version, listed in the above “reasons to reject.”